- Molecular evidence on potential contribution of
- marine emissions to aromatic and aliphatic
- organic sulfur and nitrogen aerosols in the
- South China Sea

- Yu Xu<sup>1,2</sup>, Yi-Jia Ma<sup>1,2</sup>, Ting Yang<sup>1,2</sup>, Qi-Bin Sun<sup>3,4</sup>, Yu-Chen Wang,<sup>5</sup> Lin Gui<sup>1,2</sup>, Hong-
- Wei Xiao<sup>1,2</sup>, Hao Xiao<sup>1,2</sup>, Hua-Yun Xiao<sup>1,2</sup>\*

- <sup>1</sup>School of Agriculture and Biology, Shanghai Jiao Tong University, Shanghai 200240,
- China
- <sup>2</sup>Shanghai Yangtze River Delta Eco-Environmental Change and Management
- Observation and Research Station, Ministry of Science and Technology, Ministry of
- Education, Shanghai 200240, China
- <sup>3</sup>Dongguan Meteorological Bureau, Dongguan, Guangdong, 523086, China
- <sup>4</sup>Dongguan Engineering Technology Research Center of Urban Eco-Environmental
- 16 Meteorology, GBA Academy of Meteorological Research, Dongguan, Guangdong,
- 17 523086, China
- <sup>5</sup>College of Environmental Science and Engineering, Hunan University, Changsha,
- 19 Hunan, 410082, China
- 20 \*Corresponding authors
- 21 Hua-Yun Xiao
- E-mail: xiaohuayun@sjtu.edu.cn
- Phone: +86-173-0183-7060

**Abstract:** The origins of marine aromatic and aliphatic secondary organic aerosols (SOA) remain elusive. Here, organosulfates (OSs) and nitrogen-containing organic compounds (NOCs) were measured in PM<sub>2.5</sub> collected in Sansha (the South China Sea), a region with minimal anthropogenic pollution, to investigate the potential impact of marine emissions on their formation. The proportion of aliphatic and aromatic OSs in the total OSs was significantly higher in Sansha than in other Chinese cities investigated. Biogenic OSs correlated significantly with aliphatic and aromatic OSs and NOCs. Two typical SOA tracers (C<sub>6</sub>H<sub>5</sub>O<sub>4</sub>S<sup>-</sup> and C<sub>7</sub>H<sub>7</sub>O<sub>4</sub>S<sup>-</sup>), which are formed via the atmospheric oxidation of marine benzene and toluene, were found to increase with rising chlorophyll-a and isoprene levels in surface seawater. Additionally, the impact of long-range transport and ship emissions on the abundance of OSs and NOCs was found to be insignificant. These results together with mantel test analysis suggest that marine-derived precursors may substantially contribute to the formation of aliphatic and aromatic OSs and NOCs in the Sansha region. Overall, this study provides the observation-based molecular evidence that marine biogenic emissions may play a significant role in the formation of aromatic and aliphatic SOA in the South China Sea.

41


















**Keywords:** Aerosol particles, Organosulfates, Nitrogen-containing organic 43 compounds, Marine emissions, Aliphatics and aromatics


#### 1. Introduction























Secondary organic aerosols (SOA) play a critical role in atmospheric environments and climate systems, influencing air quality, climate, elemental biogeochemical cycle, and human health (Takeuchi et al., 2022; Jimenez et al., 2009; Huang et al., 2014). While much of the previous research has primarily focused on terrestrial SOA, there has been an increasing interest in marine-derived SOA in the past decade due to their potential influence on cloud formation and radiative forcing over marine areas (Lawler et al., 2020; Wang et al., 2023; Wohl et al., 2023; Hansen et al., 2014; Yu and Li, 2021; Meskhidze and Nenes, 2006). Understanding the occurrence, origins, and formation of marine SOA is essential for accurately assessing their climatic and environmental impacts. The identification of SOA molecular markers is a fruitful approach to investigate the origins and formation processes of marine SOA (Wang et al., 2023). These markers function as chemical fingerprints, enabling researchers to trace precursors and elucidate transformation pathways (Bushrod et al., 2024; Yang et al., 2024; Jaoui et al., 2019; Ma et al., 2025). Although anthropogenic inputs transported from coastal and inland regions can contribute to the marine SOA (Zhou et al., 2023), volatile organic compounds (VOCs) emitted by phytoplankton and other marine biota are the primary origins of marine native SOA (Zhao et al., 2023; Hu et al., 2013). It has been documented that the oxidation of biogenic isoprene, monoterpenes, and dimethyl sulfide in the marine boundary layer promotes the formation of SOA markers, such as organosulfates (OSs), nitrooxy OSs, and oxygenated hydrocarbons (Hu et al., 2013; Wang et al., 2023;

Hansen et al., 2014; Moore et al., 2024). As indicated by a recent observational study, marine biogenic emissions of benzene and toluene have the potential to contribute to the formation of SOA in the oceanic environment, though direct aerosol molecular evidence for this is currently lacking (Wohl et al., 2023). Benzene and toluene are traditionally associated with anthropogenic combustion sources (Cabrera-Perez et al., 2016). These compounds have been detected in the air of temperate oceans and remote polar areas (Colomb et al., 2009; Wohl et al., 2023), indicating their widespread distribution as organic species in the marine atmosphere. Indeed, marine phytoplankton and other marine biota also have the capability of releasing various aliphatic and aromatic VOCs (Yu and Li, 2021; Weisskopf et al., 2021). However, the lower abundance of marine aliphatic and aromatic VOCs compared to isoprene and monoterpenes makes their detection challenging, potentially resulting in their oversight in atmospheric models and research endeavors (Yu and Li, 2021; Zhao et al., 2023). Specifically, aliphatic- and aromatic-derived SOA markers (e.g., aliphatic and aromatic OSs) are rarely identified in marine atmospheric observation research (mainly focusing on isoprene- and monoterpenes-derived SOA) (Wang et al., 2023). Consequently, our present understanding of the role of marine-derived aliphatic and aromatic VOCs in the formation of organic aerosols remains limited. Extant research on marine biogenic SOA (BSOA) has been concentrated on temperate and polar regions (Hawkins et al., 2010; Wozniak et al., 2014; Lawler et al., 2020; Wang et al., 2023; Bushrod et al., 2024), with relatively limited attention given























to the tropics, where environmental conditions differ significantly. The South China

Sea, with tropical climate, is one of the most biologically productive marine regions (Zhai et al., 2018), positioning it as an optimal location to study the marine BSOA components. Here, we presented the quantification of 92 OSs and measurements of nitrogen-containing organic compounds (NOCs) and other chemical components in PM<sub>2.5</sub> collected in the Sansha region (South China Sea) over a one-year period. This study aims to elucidate the potential origins and formation of tropical marine aerosol OSs and NOCs, with a focus on the impacts of marine organism emissions on the formation of aromatic and aliphatic organic aerosols. The findings of this work were expected to not only provide crucial molecular evidence for marine-derived aromatic and aliphatic SOA markers but also help to deepen our understanding of the ocean-atmosphere interaction.

### 2. Materials and methods

#### 2.1. Study site description and sample collection

The Sansha area, situated in the South China Sea, is located more than 410 kilometers from Hainan Island. This area has a tropical maritime climate, which is characterized by high average temperatures (27°C) and relative humidity (84%) throughout the year (**Table S1**). The favorable climate of the Sansha region is conducive to its rich biodiversity and complex ocean ecosystems. Aerosol sampling was conducted from August 23, 2017 to August 28, 2018 on the roof (~15 m above the ground) of the Xisha Deep Sea Marine Environment Observation and Research Station, South China Sea Institute of Oceanology (Chinese Academy of Sciences)

(**Figure S1**) in the Sansha area. For a considerable duration, Sansha has functioned as a pivotal nexus of geopolitical tensions in the South China Sea (Carrico, 2020; Ming, 2016). Consequently, the region has been in a closed or semi-closed state, particularly during the period of the 2018 US-China trade dispute. Without considering the impact of long-range transport of pollutants, the unique geographical characteristics of Sansha resulted in a considerable small anthropogenic pollution in the local area during the sampling campaign.

PM<sub>2.5</sub> samples were collected at a steady flow rate  $(1.05 \pm 0.03 \text{ m}^3 \text{ min}^{-1})$  on prebaked (500 °C for 10 h) quartz fiber filters (Pallflex, Pall Corporation, USA) using a high-volume air sampler (KC-1000, Laoying, China). The collection period for each PM<sub>2.5</sub> sample was approximately 72 h to ensure that the enriched sulfur- and nitrogencontaining organic compounds could be determined. A blank filter sample was collected on a monthly basis using the same air sampler (without operation) as referenced above. A total of 71 samples were collected and stored at -30 °C (Sect. S1). The data of relative humidity (RH), temperature (T), and the concentrations of nitrogen oxides (NO<sub>x</sub>) and ozone (O<sub>3</sub>) during the sampling campaign were obtained from the adjacent monitoring stations. These data will be averaged based on the collection time of each PM<sub>2.5</sub> sample.

#### 2.2. Measurements and data analysis

The procedures for extraction, identification, and quantification of OSs have been described in detail in our recent studies (Yang et al., 2024; Yang et al., 2023).

Briefly, the samples were extracted with methanol, followed by filtration using a 0.22 µm Teflon syringe filter (CNW Technologies GmbH). Thereafter, the solution was concentrated under a gentle stream of nitrogen gas. Finally, the extracts mixed with 300  $\mu$ L Milli-O water (~18.2 M $\Omega$  cm) were used for analysis (Acquity UPLC-MS/MS; Waters, USA). The processing of the mass spectrometry data was conducted using the MassLynx v4.1 software, which facilitated the acquisition of information pertaining to m/z, formula, and signal intensity. The identification of 33 types of OS compounds (including C<sub>6</sub>H<sub>5</sub>O<sub>4</sub>S<sup>-</sup> and C<sub>7</sub>H<sub>7</sub>O<sub>4</sub>S<sup>-</sup>) has been validated through the confirmation of sulfur-containing fragment ions (e.g., m/z 80, 81, and 96) by MS/MS analysis (Yang et al., 2023; Wang et al., 2021). More details regarding the identification, classification, and quantification of OSs and relevant limitations were described in Sect. S2. A total of up to 155 OSs were identified, of which 92 OSs were quantified using the surrogate standards. All OSs were undetectable in the blank samples with the same measurement methodology as mentioned above. The recoveries of the surrogate standards ranged from 84% to 94% (Sect. S2). It should be noted that the potential impacts of sampling without removal reactive gases (e.g., SO<sub>2</sub> and NO<sub>x</sub>) for OS and NOC (introduced below) analysis were not considered, as indicated by many previous studies (Yang et al., 2024; Wang et al., 2023; Jaoui et al., 2019; Huang et al., 2014; Ma et al., 2024). This is because if SO<sub>2</sub> and NO<sub>x</sub> can react with compounds on filters to generate SOA, analogous reaction processes can also take place on ambient particles prior to sampling. Indeed, no study has been conducted to systematically evaluate the relative efficiency of SOA formation in ambient particles and filters.






















The extraction procedure of NOCs was similar to that of OSs. NOCs were also analyzed using Acquity UPLC-MS/MS system. However, NOC identification (i.e., CHON-, CHON+, and CHN+ compounds) was conducted in both electrospray ionization positive (ESI+) and negative (ESI-) ion modes. The detailed extraction and analysis methodologies for NOCs have been well documented in our recent publications (Ma et al., 2024; Ma et al., 2025) and Sect. S2. Given the inherent uncertainties associated with ionization efficiencies across diverse compounds and the utilization of different measuring instruments (Ditto et al., 2022), an intercomparison of the relative abundance of compounds identified with the same analytical approach and instrument by the same person was performed in the present study (Ma et al., 2025). This can maximize the consistency of analysis and the reliability of the results. Furthermore, we classified the OS groups as follows (details in Sect. S2): isoprene-derived OSs (OS<sub>i</sub>), monoterpenes-derived OSs (OS<sub>m</sub>), aliphatic- and aromatic-derived OSs (they were collectively referred to as OSa), and OSs with two or three carbon atoms (i.e., C<sub>2</sub>-C<sub>3</sub> OSs) (Tables S2-S3). The system identification of potential precursors of NOCs was conducted by synthesizing the methodology delineated in previous studies (Nie et al., 2022; Guo et al., 2022; Ma et al., 2025) (Sect. S3). The categorization of CHON- and CHON+ compounds was refined into the following subgroups, including isoprene-, monoterpenes-, aromatic-, and aliphatic-derived oxidized-NOCs (Ox-NOCs) and aromatic-, aliphatic-, and heterocyclic-derived reduced-NOCs (Re-NOCs). Additionally, CHN+ compounds were categorized into monoaromatic, polyaromatic, and aliphatic subgroups. A






















comprehensive account of the modified workflow for the classification of NOCs based on potential precursors was furnished in **Figure S2** and **Sect. S2**.























For the analysis of inorganic ions, an additional portion of each filter sample was extracted with Milli-Q water using an ultrasonic bath (~4 °C). The extracts were subsequently filtered through a PTFE syringe filter. Inorganic ions including SO<sub>4</sub><sup>2-</sup>, NH<sub>4</sub><sup>+</sup>, K<sup>+</sup>, NO<sub>3</sub><sup>-</sup>, Mg<sup>2+</sup>, Ca<sup>2+</sup>, Na<sup>+</sup>, and Cl<sup>-</sup> were quantified using an ion chromatograph system (Dionex Aquion, Thermo Scientific, USA) (Xu et al., 2019; Xu et al., 2022; Lin et al., 2023). The pH value was predicted with a thermodynamic model (ISORROPIA-II) (Xu et al., 2020; Xu et al., 2023; Yang et al., 2024; Xu et al., 2022) (details in Sect. S4). The sea surface temperature (SST) data (0.25°×0.25°) were derived from NOAA Optimum Interpolation Sea Surface Temperature Version 2 (OISSTv2) dataset. Chlorophyll-a (Chl a) concentration data (0.5°×0.5°) were derived from the MODIS Level-3 product processed with the OC5 algorithm and acquired from the GlobColour. To prevent the extraction of missing values, the nearest available values within the radius of 0.75° and 1° around the sampling location were averaged and assigned. Isoprene concentrations in surface seawater were predicted by an empirical formulas that depends on temperature and Chl a (i.e., 20.9[Chl a] -1.92[SST] + 63.1) (Wang et al., 2023). Notably, although uncertainties may exist in predictions of seawater-surface isoprene, the predicted trend in its concentration changes was expected to be reliable. The gridded VOC emissions from international shipping (0.1°×0.1°) were extracted from the HTAPv3 mosaic emission inventory (Crippa et al., 2023; Huang et al., 2017). To characterize the long-range transport pathways of air masses arriving at the sampling site during each sampling event, 3day (72 h) back trajectories beginning at 500 m above sea level were computed using TrajStat v1.5.4 implemented in MeteoInfoMap 3.3.0 (Chinese Academy of Meteorological Sciences, China). The data used for trajectory calculation were Air obtained from NOAA's Resources Laboratory (ftp://arlftp.arlhq.noaa.gov/pub/archives/gdas1/). The fire data were derived from Fire Information for Resource Management NASA's System (https://firms.modaps.eosdis.nasa.gov/active\_fire/). The concentrations of non-sea-salt (nss) potassium ion was calculated by subtracting 0.022 times the sodium ion concentration from the total potassium ion concentration (Chen et al., 2010).

211

212

213











#### 3. Results and discussion

## 3.1. Compositions and abundances of OSs and NOCs

Figure 1a compares the average concentrations of OS<sub>i</sub>, OS<sub>m</sub>, C<sub>2</sub>-C<sub>3</sub> OSs, and 214 OS<sub>a</sub> in Sansha with those observed in other regions worldwide. OS<sub>i</sub> was the most 215 abundant aerosol OS subgroup in Sansha, averaging 12 ± 3 ng m<sup>-3</sup> and contributing 216  $32 \pm 3\%$  to the total OSs. The second most abundant OS subgroup was  $C_2$ – $C_3$  OSs 217  $(30 \pm 4\%)$ , followed by OS<sub>a</sub>  $(23 \pm 2\%)$  and OS<sub>m</sub>  $(16 \pm 2\%)$ . The total OS 218 concentrations in Sansha ranged from 26 to 93 ng m<sup>-3</sup>, exhibiting an average of 39 ± 219 13 ng m<sup>-3</sup>. A global summary of the observation data suggested that the highest mean 220 OS concentration (2157 ng m<sup>-3</sup>) was recorded at a forest park site in Look Rock, TN, 221 USA (Budisulistiorini et al., 2015) and the lower level (generally less than 2 ng m<sup>-3</sup>) 222

was observed in both urban and polar sites (Hansen et al., 2014; Ma et al., 2014). The observed OS concentrations in this study fell within the range of values previously documented (**Table S4**). Generally, the predominance of OS<sub>i</sub> in the total OSs has been extensively reported in various observations conducted in different regions, including Xi'an, China (Yang et al., 2025), Guangzhou, China (Bryant et al., 2021; Yang et al., 2025), Beijing, China (Yang et al., 2025), Shanghai, China (Yang et al., 2023), the Yellow Sea, China, Patra, Greece (Kanellopoulos et al., 2022), Towson, MD, USA (Meade et al., 2016), and the Amazon forest (Riva et al., 2019) (**Table S4**). A plausible explanation for these observations is the presence of a substantial biogenic emission of isoprene.

**Figure 1**. A comparison of (a) the concentrations of isoprene-derived OSs (OS<sub>i</sub>), monoterpenes-derived OSs (OS<sub>m</sub>), C<sub>2</sub>–C<sub>3</sub> OSs, and aliphatic- and aromatic-derived OSs (OS<sub>a</sub>) in PM<sub>2.5</sub> in Sansha and other regions worldwide (more data in **Table S4**). (b) Average signal intensity distributions for the CHON+, CHN+, and CHON-

compounds from various sources in PM<sub>2.5</sub> collected in Sansha and other Chinese cities. The NOC data were identified using the identical analysis methodology (Ma et al., 2025). In addition, it is important to acknowledge that not all OS species have been detected in all locations, and the absence of OSs in some panels does not necessarily imply their nonexistence. The figure may contain a territory that is disputed according to the United Nations.

















A notable finding of this study was that the proportion of OSa in the total OSs in Sansha Island (23%, on average) was higher than the observed results in most other cities (Figure 1a and Table S4). In order to minimize potential discrepancies stemming from different measurement methods and the types of OS detected, we conducted a comparison of our previous OS data, measured in different regions of China, with the data obtained from Sansha (Yang et al., 2023; Yang et al., 2025). These data were evaluated using the identical analysis methodology. Indeed, the mass fractions of OS<sub>a</sub> in the total OSs in Xi'an (inland; 19%), Guangzhou (coastal; 18%), and Shanghai (coastal; 12%) were smaller than that in Sansha Island, although the OS<sub>a</sub> concentration was lower in Sansha Island (Figure 1a and Table S4). The attribution of this phenomenon to the long-range transport of pollutants from coastal cities in China may not be a reasonable assumption. This is due to the fact that the transmission effect was incapable of inducing an increase in the proportion of OSa from coastal cities with developed industries (e.g., Shanghai and Guangzhou) to Sansha Island (see section 3.3 for further evidence). Despite the absence of quantitative research on the aliphatic and aromatic OSs in the marine aerosols previously, the relatively high proportion of  $OS_a$  in the Sansha region compared to industrialized inland and coastal cities in China suggests either marine phytoplankton (/microorganisms) or ships had released large amounts of aliphatic and aromatic precursors to promote the formation of  $OS_a$  locally.























Figure 1b presents the average signal intensity distributions of different NOCs from various precursors in Sansha, inland (Beijing and Haerbin; urban site) (Ma et al., 2025), and coastal (Hangzhou; urban site) (Ma et al., 2025) areas of China. On average, aromatic-derived Ox-NOCs accounted for 12% of the total signal intensity of CHON- compounds in Sansha, which was significantly lower than the observations (72%–90%) recorded in Beijing, Haerbin, and Hangzhou. However, the mean signal intensity proportion of identified isoprene- and monoterpenes-derived Ox-NOCs in the total signal intensity of CHON- compounds was much higher in Sansha (29%) than in Beijing, Haerbin, and Hangzhou (2%–3%). The mean atmospheric NO<sub>2</sub> level in the Sansha area (Table S1) was found to be more than 10 times lower than that observed in inland urban areas (Ma et al., 2025), which may significantly constrain the formation of aromatic-derived Ox-NOCs. However, the total release of isoprene and monoterpenes from marine sources in the Sansha area may account for a higher proportion of the total NOC precursors than aromatic compounds, likely causing the proportion of isoprene- and monoterpenes-derived Ox-NOCs to be relatively higher than that of aromatics-derived Ox-NOCs.

For Re-NOCs, the mean signal intensity proportion of aromatic-derived CHON+

in the total signal intensity of CHON+ compounds was also lower in Sansha (30%) than in Beijing (48%), Haerbin (88%), and Hangzhou (35%). The reaction of carbonyl compounds with NH<sub>4</sub><sup>+</sup> (or NH<sub>3</sub>) in the aqueous phase has been suggested to form Re-NOCs (Gen et al., 2018). The Sansha region exhibits high RH (Table S1), which provides an aerosol environment conducive to the aqueous-phase reactions (Xu et al., 2023; Ma et al., 2025; Yang et al., 2024; Xu et al., 2020). Presumably, the significantly lower levels (up to ten times lower) of aerosol NH<sub>4</sub><sup>+</sup> (**Table S1**) and gaseous NH<sub>3</sub> in the Sansha area relative to other cities (Ma et al., 2025; Pan et al., 2018; Dong et al., 2023) may be one of the important factors constraining Re-NOC formation. The insignificant correlation between CHON+ abundance and NH4+ concentration (P > 0.05) also partially supports the above inference. Another potential factor contributing to the lower abundance of aromatic-derived Re-NOCs in the Sansha area could be the substantially reduced levels of their precursors, which is a consequence of the minimal anthropogenic pollution there compared to other urban regions. In addition, the signal intensity fraction of aromatic-derived CHN+ type Re-NOCs in the total signal intensity of CHN+ compounds was lower in Sansha (28%) than in inland urban pollution areas (37%–41%; Beijing and Haerbin) but higher than in the coastal area (14%; Hangzhou). Clearly, the formation of aerosol aromaticderived NOCs in the Sansha area may be hindered by inadequate local availability of NO<sub>x</sub>, NH<sub>4</sub><sup>+</sup>, or VOC precursors. A similar spatial variation pattern was observed in the peak intensity fractions of aliphatic-derived Ox-NOCs and aromatic-derived Re-NOCs (Figure 1). In contrast, the peak intensity fraction of aliphatic-derived Re-






















NOCs in the Sansha area fell within the ranges observed in other cities. Owing to the abundant source of marine sulfate and the strong solar radiation in the tropical ocean (Sansha) (Xiao et al., 2017), sulfate availability and atmospheric oxidation capacity would not be limiting factors for OS formation. Thus, although the abundance proportions of aromatic-derived NOCs and aliphatic-derived Ox-NOCs were lower in the Sansha area than in the inland cities (in contrast to OSs, **Figure 1**), the overall observation results at least suggest that the composition of organic aerosols (particularly OSs) in this region has been significantly influenced by aliphatic and aromatic precursors. The subsequent discussions will provide further evidence to support the above consideration.

#### 3.2. Temporal variations of OSs and NOCs and implications for their origins

Figures 2a,b show the temporal variations in the concentrations of the various OS subgroups. The mass concentrations of  $OS_i$ ,  $OS_m$ , and  $OS_a$  tended to increase from autumn (Aug – Nov) to winter (Dec – Feb), with the maximum value in February. Subsequent to February, there was a gradual decrease in the concentrations of various OSs, and the fluctuations in their concentrations over time stabilized. It should be noted that this tropical sea area does not exhibit a distinct seasonal pattern, attributable to the small air temperature variation between summer and winter (Table S1). Additionally, the SST value was lowest in February ( $24 \pm 1$ °C) (Figure S3a and Figure S4), corresponding to the highest levels of isoprene in surface seawater (Figure S3b). The isoprene concentration in surface seawater underwent a decrease

and subsequent stabilization after April, exhibiting a resemblance to the temporal patterns observed in various OSs (Figure 2 and Figure S3). The variation patterns among SST, seawater isoprene levels, and Chlorophyll-a obtained from empirical formulas or satellite data presented here are consistent with actual observation results in the South China Sea (Zhai et al., 2018). It is therefore evident that the parameters applied here (e.g., seawater isoprene levels and Chlorophyll-a) function as reliable indicators. These findings suggest that the formation of these OSs, at least for well-defined biogenic VOCs-derived OSs (e.g., OS<sub>i</sub> and OS<sub>m</sub>), may be closely related to marine phytoplankton emissions.

Sulfate and particle acidity have been suggested to be the important factors controlling the formation of OSs (Yang et al., 2023; Surratt et al., 2008; Brüggemann et al., 2020). Due to the high sulfate level and low ammonium level in PM<sub>2.5</sub> in the Sansha area, the investigated aerosol particles are all acidic (**Table S2**). This is favorable for aerosol OS formation. The aerosol sulfate concentration in the Sansha area was also higher than our observed value in Shanghai, while the various OS concentrations were found to be lower than those in Shanghai (the same OS measurement method and identification type) (Yang et al., 2023). Furthermore, insignificant correlations (P > 0.05) were found between sulfates and OS<sub>i</sub>, OS<sub>m</sub>, C<sub>2</sub>–C<sub>3</sub> OSs, and OS<sub>a</sub>, as expected from the dissimilar temporal patterns for these variables (**Figures 2a,b,e**). Interestingly, these OSs showed significant correlations (P > 0.6, P < 0.01) with local O<sub>3</sub> levels. The above results suggest that aerosol OSs in the Sansha area may be mainly formed locally and are tightly associated with

precursor emission levels (e.g., abovementioned phytoplankton emissions) rather than sulfates. Aerosol OS<sub>a</sub> is generally regarded as anthropogenic-derived OSs in inland urban areas (Yang et al., 2024; Yang et al., 2025). Previous studies have suggested that marine phytoplankton and other marine biota are capable of releasing aliphatic and aromatic VOCs (Yu and Li, 2021; Weisskopf et al., 2021). Thus, the above findings may also imply that aromatic and aliphatic precursors emitted by marine organisms are potential sources of aerosol OS<sub>a</sub> in the Sansha area.

Figure 2. Temporal variations in (a, b) the concentration of the different OSs, (c-f) the signal intensity of the various CHON+, CHN+, and CHON- compounds, and (f) the concentration of sulfate and ozone. Ali\_Re\_NOCs (+), Aro\_Ox\_NOCs (+), and Ali\_Ox\_NOCs (+) refer to the aliphatic-derived reduced form NOCs, aromatic-derived oxidized form NOCs, and aliphatic-derived oxidized form NOCs identified in ESI+ mode, respectively. Aro\_Re\_NOCs (-), Aro\_Ox\_NOCs (-), and Ali\_Ox\_NOCs (-) refer to the aromatic-derived reduced form NOCs, aromatic-derived oxidized form

NOCs, and aliphatic-derived oxidized form NOCs identified in ESI- mode, respectively.


















363

364

Figures 2c,d show the temporal variations in the signal intensities of the various aromatic- and aliphatic-derived NOCs. These aromatic- and aliphatic-derived NOCs identified in ESI+ mode showed an overall consistent temporal trend, a pattern of which was highly similar to that of  $OS_i$  and  $OS_m$  (r up to 0.82, P < 0.01). Moreover, the aromatic- and aliphatic-derived NOCs detected in ESI+ mode were significantly (0.55 < r < 0.66, P < 0.01) positively correlated with O<sub>3</sub> and O<sub>x</sub> (O<sub>3</sub> + NO<sub>2</sub>). In contrast, the aromatic- and aliphatic-derived NOCs detected in ESI- mode showed a relatively weak positive correlation (0.36 < r < 0.49) with O<sub>3</sub> and O<sub>x</sub>. OS<sub>i</sub> and OS<sub>m</sub> were two typical types of biogenic OSs (Yang et al., 2023; Wang et al., 2023; Surratt et al., 2008). Thus, the overall results not only suggest that the formation of OSa was related to marine biogenic sources (as discussed above), but also imply that the formation of aromatic and aliphatic NOCs (at least for those detected in ESI+ mode) may be influenced by biogenic sources. The subsequent discussion will present additional evidence regarding the impact of aromatic and aliphatic precursors released from the ocean on the formation of aerosol sulfur- and nitrogen-containing organic compounds.



## 3.3. Insignificant impact of long-range transport and ship emissions

The long-range transport of pollutants from coastal areas or marine traffic (i.e., ship) emissions has been suggested to be an important contributor to the inorganic and organic components of marine aerosols (Zhou et al., 2023; Wang et al., 2023). Specifically, a previous study in the Sansha area during the years 2014–2015 suggested that the inorganic components in aerosols may be influenced by biomass burning-related pollutants transported from the northeast and southwest directions of Sansha (Xiao et al., 2017). To examine the potential roles of long-range transport and ship emissions in the formation of aerosol OSs and NOCs in the Sansha area, variations in the air mass transport patterns (Figure 3a), nss-K<sup>+</sup> abundances (Figure 3b), and ship emissions of non-methane VOCs were investigated (Figure 4). Cluster analysis of air mass backward trajectories indicated that air masses arriving at the sampling site primarily originated from the northeast sea area during the winter and spring months (December - May). This period coincided with heightened biomass burning activity in the southwest coastal regions (Figure 3). Conversely, during other periods, biomass burning activity exhibited a marked decline, with air masses predominantly originating from the southwestern and northeastern sea areas. Nevertheless, the concentrations of nss-K<sup>+</sup>, used as a typical biomass burning tracer (Liang et al., 2021), demonstrated no variability in accordance with the direction of the air masses or the intensity of biomass burning in coastal areas. For example, a relatively stable level of nss-K<sup>+</sup> was observed during the specified period (December - May), which corresponded to the period of maximum biomass burning intensity. The high nss-K<sup>+</sup> event occurred in August 2018, which can be attributed to 94% of the






















air masses passing through coastal areas where biomass burning occurred. However, the OS concentrations did not increase during that month. In particular, the Sansha area belongs to the tropical ocean with high ultraviolet radiation, making it almost impossible for VOCs such as isoprene and gaseous reactive nitrogen to be transported here from the nearest coast more than 410 kilometers away. Even if we assume that biomass burning in inland or coastal areas transported OSs to the Sansha region, it would only lead to a significant increase in OSa abundances but not induce a synchronous increase in OSi, OSm, and OSa abundances (Figure 2). These results indicate that long-range transport exerted a weak impact on organic aerosol composition in the Sansha area.

**Figure 3.** (a) Air mass backward trajectories (72 h; 500 m above ground level) and fire spots (NASA active fire data, VIIRS 375 m) during different periods, with corresponding temporal variations in (b) the concentration of nss-K<sup>+</sup>. The map in panel (a) was derived from MeteoInfoMap (Chinese Academy of Meteorological Sciences, China).

**Figure 4** Illustrations presenting (a) the variation in the amount of non-methane VOCs (NMVOCs; with the exception of isoprene and monoterpenes) emitted from ships over months in the Sansha area. (b) Temporal variations in the emission amount of several important NMVOCs. The small and large red circles in panel b show radii of 0.25° and 0.75°, respectively.

Figures 4a,b show the temporal variation in the amount of the total non-methane VOCs (NMVOCs; with the exception of isoprene and monoterpenes) and major VOCs emitted from ships in the Sansha area. The emission levels of NMVOCs, isoprene, and monoterpenes were lowest in February–March (Figure 4b and Figure S5), which was opposite to the temporal variation patterns of various OSs and aromatic- and aliphatic-derived NOCs (Figure 2). In other months, the level of NMVOCs from ship emissions remained relatively stable. This can be attributed to the unique geographical location of the Sansha area, where the government exercises stringent control over the number of passable ships in the surrounding sea areas. Thus,

the above results suggest that ship emissions were not the primary factor in controlling the abundance of aerosol OSs and NOCs in the Sansha area. These findings, combined with the previously mentioned fact that OSa proportion was higher in Sansha than in China's industrialized inland and coastal cities, further demonstrate that the ocean may release aliphatic and aromatic precursors to facilitate in situ OSa formation.

# 3.4. Potential impact of marine biological emissions to aromatic and aliphatic

## organic sulfur and nitrogen aerosols

According to the aforementioned analysis, we can propose that the aerosol OSs and aromatic- and aliphatic-derived NOCs in the Sansha area may be locally formed and closely related to the marine biological emissions. A Mantel test correlation analysis was conducted to further investigate the impact of different factors on the formation of OSs and NOCs in the Sansha area (Figure 5). We found that the abundances of OS<sub>i</sub>, OS<sub>m</sub>, and OS<sub>a</sub> were significantly related to various impacting factors, including Na<sup>+</sup>, RH, O<sub>3</sub>, O<sub>x</sub>, Chl a, and SST. Sodium ions and Chl a have been identified as important indicators of marine sources (Wang et al., 2023; Xiao et al., 2017). Ozone and its photolysis related products (major source of ·OH in the ocean atmosphere) are important oxidants for OS formation (Yang et al., 2023). The RH of the air in the Sansha area is consistently high throughout the year (84%, Table S1), which plays a significant role in the liquid phase formation of OSs (Yang et al., 2024).

control the formation of  $OS_i$  and  $OS_m$ , but also significantly affect the formation of aliphatic- and aromatic-derived OSs. In addition, the principal component analysis result showed a high aggregation effect of  $Na^+$ ,  $O_3$ ,  $O_x$ , Chl a with various OSs (**Figure 6a**), further indicating that the emissions of marine phytoplankton (/microorganisms) may significantly contribute to the local formation of aerosol OSs in the Sansha area.

**Figure 5.** Mantel test correlation heatmap showing the interrelationships between different factors within the same matrix on the left-hand side and the correlation of different matrices on the right-hand side. The size of the solid square indicates the significance of the correlation between the two corresponding parameters. The larger square indicates that the correlation is more significant. The colors of the different

solid circles indicate different correlation coefficients (*r*). OS<sub>b</sub> comprises OS<sub>i</sub> and OS<sub>m</sub>. NOC\_b incorporates isoprene- and monoterpenes-derived NOCs. NOC\_aro\_h includes Aro\_Re\_NOCs, Aro\_Ox\_NOCs, and heterocyclic-derived NOCs. NOC\_ali includes Ali\_Re\_NOCs and Ali\_Ox\_NOCs. The '+'and '-' symbols in parentheses refer to the NOCs identified in ESI+ and ESI- modes, respectively.

**Figure 6.** Principal component analysis result (a) deciphering the interrelationships among various OSs, various NOCs, and key factors that likely influence their formation. Diagrams presenting (b) the Spearman correlations between benzene- and toluene-derived OSs and indicators of ocean sources. The colors of the different solid circles indicate different correlation coefficients (r), with specific values labeled in the circles. The larger circle indicates that the correlation is more significant, while the symbol "×" indicates that the P value is greater than 0.05. The illustration in the

bottom of panel (b) shows that  $C_6H_5O_4S^-$  and  $C_7H_7O_4S^-$  can be derived from the transformation of benzene and toluene emitted from the ocean.





















488

489

In addition, we found that aliphatic-, aromatic-, and heterocyclic-derived NOCs identified in ESI+ mode were significantly correlated with either Na<sup>+</sup> or O<sub>3</sub> and O<sub>x</sub> (Figure 5). The various NOCs derived from aliphatic, aromatic, and heterocyclic precursor and the factors including Na<sup>+</sup>, O<sub>3</sub>, O<sub>x</sub>, and Chl a were also characterized by a high degree of aggregation in the plot of the principal component analysis (Figure 6a). These results provide additional evidence that the abundance of these aromatic and aliphatic NOCs was likely influenced by marine emissions and atmospheric secondary processes. It is important to acknowledge that the levels of atmospheric NO<sub>2</sub> (5  $\pm$  6  $\mu$ g m<sup>-3</sup>) and aerosol NO<sub>3</sub><sup>-</sup> (0.2  $\pm$  0.2  $\mu$ g m<sup>-3</sup>) and NH<sub>4</sub><sup>+</sup> (0.7  $\pm$  0.7  $\mu$ g m<sup>-3</sup>) in the Sansha area (Table S1) were considerably lower compared to those observed in urban areas (Xu et al., 2024). The presence of low levels of inorganic nitrogen components posed a substantial restriction on the formation of NOCs. However, these low-level inorganic nitrogen compounds are not the main factors that constrain the formation of OSs. Consequently, the identification of potential sources (associated with marine phytoplankton and other marine biota) contributing to OS formation is more indicative of the impact of marine emissions on isoprene-, monoterpenes-, aliphatic-, and aromatic-derived SOA in the Sansha area.

The release of benzene and toluene from marine organisms has been suggested to be important precursors for marine SOA (Wohl et al., 2023). It has been established

that the reaction of benzene and toluene with sulfate radicals in the aqueous phase can lead to the formation of two aromatic OSs, namely C<sub>6</sub>H<sub>5</sub>O<sub>4</sub>S<sup>-</sup> and C<sub>7</sub>H<sub>7</sub>O<sub>4</sub>S<sup>-</sup> (Huang et al., 2020). We have detected these two aromatic OSs in aerosols collected in the Sansha area, the structures of which were shown in **Figure 6b**. Moreover, we found that these two benzene- and toluene-derived OSs showed significant positive correlations with multiple indicators of marine emissions (e.g., Chl a, surface seawater isoprene, Na<sup>+</sup>, OS<sub>i</sub>, and OS<sub>m</sub>) (**Figure 6b**). This further corroborates the important effect of marine emissions on the formation of aromatic-derived SOA in the Sansha area.

## 4. Conclusion and atmospheric implications

Based on our current understanding, this study represents the inaugural instance of simultaneous comprehensive characterization of OSs and NOCs (in both ESI+ and ESI- modes) in PM<sub>2.5</sub> in tropical marine areas, particularly in the Sansha area with minimal anthropogenic pollution. We concluded that the emissions of marine organisms can contribute to the formation of both typical BSOA (i.e., isoprene and monoterpenes-derived species) and aliphatic- and aromatic-derived SOA in this sea area. A recent study in the Yellow Sea of China has reported that marine phytoplankton or microorganisms can contribute to the formation of marine aerosol  $OS_i$  and  $OS_m$  (Wang et al., 2023). The systematic analysis of this study, including the significant correlation (r = 0.61-0.71, P < 0.01) between estimated surface seawater isoprene and  $OS_i$  and  $OS_m$ , can indeed directly demonstrate the important role of

marine biological emissions in the formation of marine aerosol BSOA. However, the available information regarding the origins of aliphatic and aromatic OSs and NOCs in marine aerosols was previously inadequate. Aerosol aliphatic and aromatic pollutants in marine areas are usually considered to be transported over long distances from land (Zhou et al., 2023; Sun et al., 2023; Hansen et al., 2014). Thus, this is the first field observation case to demonstrate that marine organisms may also provide important aliphatic or aromatic precursors for the formation of aliphatic and aromatic OSs and NOCs.

The presence of benzene and toluene has been observed in the marine and polar atmosphere (Colomb et al., 2009; Wohl et al., 2023). Recent observational evidence has indicated that the emission of benzene and toluene from marine organisms contributes to the formation of marine SOA (Wohl et al., 2023). All of these studies underscored the potential of marine organisms to become important contributors to the formation of aliphatic and aromatic OSs and NOCs. If we could directly measure various types of aliphatic and aromatic VOCs on the surface of near sea water in remote ocean areas (without anthropogenic pollution), this would undoubtedly provide the most direct evidence of the aforementioned considerations. However, direct detection is challenging due to the low environmental levels of those VOCs in remote ocean areas (low seawater Chl a levels). Furthermore, low levels of marine organisms-derived aliphatic and aromatic VOCs also pose a challenge for directly measuring their characteristic products (e.g., OSs) after oxidation. Thus, the approach of collecting atmospheric fine particles over an extended period (e.g., 23h – 168h)

(Wang et al., 2023; Hansen et al., 2014; Miyazaki et al., 2016; Lawler et al., 2020) that we adopt is a highly desirable strategy to maximize the enrichment of OA components in the sea region. Future studies will necessitate the use of ultra-high resolution mass spectrometry combined with specially designed loading equipment for on-line detection of various trace aliphatic and aromatic VOCs released from the oceans and their fluxes. In particular, the generation of SOA on the ocean surface from these aliphatic and aromatic precursors also requires systematic mechanistic studies.

**Data availability.** The data presented in this work are available upon request from the corresponding authors.

**Supplement.** The supplement related to this article is available online at...

**Author contributions.** HYX and YX designed the study. TY, YJM, QBS, LG, HWX, and HX performed field measurements and sample collection; TY and YJM performed chemical analysis; YX performed data analysis; YX wrote the original manuscript; HYX and YCW provided suggestions, and YX reviewed and edited the manuscript.

**Acknowledgements.** The authors are very grateful to the editor and four anonymous referees for their kind and valuable comments, which improved the paper.

Financial support. This study was kindly supported by Key Program of the National Natural Science Foundation of China (grant number 42430501), National Natural Science Foundation of China (grant number 42303081), National Key Research and Development Program of China (grant number 2023YFF0806001), and Shanghai "Science and Technology Innovation Action Plan" Shanghai Sailing Program through grant 22YF1418700.

Conflict of Interest. The authors declare no conflicts of interest relevant to this study.

Review statement. This paper was edited by Dara Salcedo and reviewed by four anonymous referees.

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
