# Peer review of "Molecular evidence on potential contribution of"

_EGUsphere, 2025_

## Author Comment (AC1)

**General.**

We would like to appreciate the referee for providing the valuable comments to improve the manuscript. We have revised our manuscript by fully taking the editor's comments into account. Responses to specific comments are described below. All the changes made and appeared in the revised text are shown in red. All detailed answers to comments are displayed in blue.

**Comments of Referee #1 and our responses to them**

*This manuscript by Xu et al. presents one-year ambient measurements on organosulfates and nitrogen-containing organic compounds in PM2.5 collected from an isolated Sansha island in the South China Sea. To my knowledge, this work represents the first concurrent field observation resolving molecular signatures of organic sulfur and organic nitrogen aerosols in Sansha Island. The authors found that the proportion of aliphatic and aromatic organosulfates in the total organosulfates was significantly higher in Sansha than in other Chinese cities investigated. This is a very interesting finding. Furthermore, the authors demonstrated that marine biogenic sources may play an important role for the production of aliphatic/aromatic organosulfur compounds and nitrogen-containing organic compounds, with relatively little from long-range transport and shipping-derived emissions. I believe this may be a special observation case to demonstrate that marine organisms can provide important aliphatic or aromatic precursors for the formation of aliphatic and aromatic organosulfates and nitrogen-containing organic compounds in the tropical ocean atmosphere.*

*In general, the manuscript is well-written and logically structured, and it presents a*

*wealth of valuable observational data. I have only a few minor suggestions, and I believe it would be suitable for publication in ACP once these suggestions are addressed.*

Response: We deeply appreciate your valuable suggestions and time spent reviewing our manuscript.

Specific comments:

*1. I can understand directly comparing the quantitative organosulfate concentrations between Sansha and other Chinese cities. However, I also found that the authors made a comparison between the signal intensity of various nitrogen-containing organic compounds in Sansha and the results from previous field studies. Due to different instruments or methods used in the determination of nitrogen containing organic compounds in different studies, there may be differences in the types and ionization efficiencies of nitrogen containing organic compound measured. Therefore, how did the author correct or explain these issues.*

Response: The NOC data used for comparison were derived from our research group (Ma et al., 2025), and all NOC data showing in Figure 1 were obtained with the same instruments, methodology, and operators.

Lines 164–166: …an intercomparison of the relative abundance of compounds identified with the same analytical approach and instrument by the same person was performed in the present study (Ma et al., 2025)…

Lines 239–240: …The NOC data were identified using the identical analysis methodology (Ma et al., 2025)…

*2. The authors present a large amount of satellite-based data. I understand that conducting year-long observations on such data in field sites with underdeveloped infrastructure is quite challenging. Therefore, I suggest that the authors incorporate more literature on observational studies carried out in the South China Sea to validate the reliability of the trends in chlorophyll-a or sea surface temperature calculated using satellite data. Since the various analytical results of this study were closely linked to the trends of the parameters, it would suffice to demonstrate the reliability of the trend changes in the key parameters.*

Response: We greatly appreciate your suggestions. The variation patterns among sea surface temperatures and Chlorophyll-a obtained from satellite data presented here are consistent with actual observation results in the South China Sea (Zhai et al., 2018).

Lines 326–329: The variation patterns among SST, seawater isoprene levels, and Chlorophyll-a obtained from empirical formulas or satellite data presented here are consistent with actual observation results in the South China Sea (Zhai et al., 2018).

*3. This study quantified up to 92 types of organosulfates. However, the authors utilized surrogate standards. I recognize that reference standards for many organic sulfur compounds are difficult to obtain or prepare, which is why numerous studies rely on surrogate standards for quantifying organosulfates. Therefore, I suggest the authors clarify the limitations of this methodology.*

Response: We have added more descriptions in the SI to clarify the limitations of OS quantification methodology.

Text S2:

Due to the absence of authentic standards, the majority of the identified OSs were quantified using surrogate standards (Hettiyadura et al., 2019; Bryant et al., 2021; Wang et al., 2018; Kanellopoulos et al., 2022; Yang et al., 2023)…

…It is imperative to acknowledge the limitations of the OS species quantified in this study. While the quantified OS concentration value should not be regarded as an accurate measurement of OS compounds, it is a best solution in the absence of authentic OS standards…

*4. Lines 285-286: '...aromatic-derived CHON+ type Re NOCs...' I suggest changing it to 'For Re NOCs, the mean signal intensity proportion of aromatic-derived CHON+ in the total signal intensity of CHON+ compounds ...'*

Response: We have updated the relevant content in the revised manuscript.

Lines 281–283: …For Re-NOCs, the mean signal intensity proportion of aromatic-derived CHON+ in the total signal intensity of CHON+ compounds was also lower in Sansha (30%) than in Beijing (48%), Haerbin (88%), and Hangzhou (35%)...

*5. Lines 295-297: Please reorganize this sentence to enhance the logical coherence of the context.*

Response: The relevant content has been rephrased in the revised manuscript (Lines

287–292).

*6. Line 302: This should be 'The subsequent discussions…'*

Response: The revision has been made (Line 312).

*7. Line 305: …Temporal variations of OSs and NOCs…*

Response: The revision has been made (Line 314).

*8. Line 329: Please delete 'very'..*

Response: The revision has been made (Line 338).

*9. Line 350: Please delete 'from different sources'.*

Response: The revision has been made (Line 356).

*10. Has the structure of the two important organosulfate markers in Figure 6 been verified?*

Response: The structure of those two important organosulfate markers in Figure 6 been verified in our previous publication (Yang et al., 2023) (Lines 141–144).

**Once again, we deeply appreciate the time and effort you've spent in reviewing our manuscript.**

**References**

Bryant, D. J., Elzein, A., Newland, M., White, E., Swift, S., Watkins, A., Deng, W., Song, W., Wang, S., Zhang, Y., Wang, X., Rickard, A. R., and Hamilton, J. F.: Importance of Oxidants and Temperature in the Formation of Biogenic Organosulfates and Nitrooxy Organosulfates, ACS Earth Space Chem., 5, 2291-2306, 10.1021/acsearthspacechem.1c00204, 2021.

Hettiyadura, A. P. S., Al-Naiema, I. M., Hughes, D. D., Fang, T., and Stone, E. A.: Organosulfates in Atlanta, Georgia: anthropogenic influences on biogenic secondary organic aerosol formation, Atmos. Chem. Phys., 19, 3191-3206, 10.5194/acp-19-3191-2019, 2019.

Kanellopoulos, P. G., Kotsaki, S. P., Chrysochou, E., Koukoulakis, K., Zacharopoulos, N., Philippopoulos, A., and Bakeas, E.: PM2.5-bound organosulfates in two Eastern Mediterranean cities: The dominance of isoprene organosulfates, Chemosphere, 297, 134103, https://doi.org/10.1016/j.chemosphere.2022.134103, 2022.

Ma, Y. J., Xu, Y., Yang, T., Gui, L., Xiao, H. W., Xiao, H., and Xiao, H. Y.: The critical role of aqueous-phase processes in aromatic-derived nitrogen-containing organic aerosol formation in cities with different energy consumption patterns, Atmos. Chem. Phys., 25, 2763-2780, 10.5194/acp-25-2763-2025, 2025.

Wang, Y., Hu, M., Guo, S., Wang, Y., Zheng, J., Yang, Y., Zhu, W., Tang, R., Li, X., Liu, Y.,

Le Breton, M., Du, Z., Shang, D., Wu, Y., Wu, Z., Song, Y., Lou, S., Hallquist, M., and Yu, J.: The secondary formation of organosulfates under interactions between biogenic emissions and anthropogenic pollutants in summer in Beijing, Atmos. Chem. Phys., 18, 10693-10713, 10.5194/acp-18-10693-2018, 2018.

Yang, T., Xu, Y., Ye, Q., Ma, Y. J., Wang, Y. C., Yu, J. Z., Duan, Y. S., Li, C. X., Xiao, H. W., Li, Z. Y., Zhao, Y., and Xiao, H. Y.: Spatial and diurnal variations of aerosol organosulfates in summertime Shanghai, China: potential influence of photochemical processes and anthropogenic sulfate pollution, Atmos. Chem. Phys., 23, 13433-13450, 10.5194/acp-23-13433-2023, 2023.

Zhai, X., Zhang, H.-H., Yang, G.-P., Li, J.-L., and Yuan, D.: Distribution and sea-air fluxes of biogenic gases and relationships with phytoplankton and nutrients in the central basin of the South China Sea during summer, Mar. Chem., 200, 33-44, https://doi.org/10.1016/j.marchem.2018.01.009, 2018.

---

## Author Comment (AC2)

**General.**

We would like to appreciate the referee for providing the valuable comments to improve the manuscript. We have revised our manuscript by fully taking the editor's comments into account. Responses to specific comments are described below. All the changes made and appeared in the revised text are shown in red. All detailed answers to comments are displayed in blue.

**Comments of Referee #2 and our responses to them**

*This manuscript presented the quantification of OSs and measurements of nitrogen-containing organic compounds (NOCs) in PM2.5 collected in the Sansha region over a one-year period. The proportion of aliphatic and aromatic OSs in the total OSs was significantly higher in Sansha than in other Chinese cities investigated. They concluded that the emissions of marine organisms can contribute to the formation of both typical BSOA and aliphatic- and aromatic-derived SOA in this sea area with few anthropogenic sources of pollution. This study can provide a scientific bases for studying on the potential origins and formation of tropical marine aerosol OSs and NOCs.*

*The manuscript is mostly well written and easy to read. The results and discussion are detailed and convincing through the comparative analysis of results. The manuscript deserves publication after the authors take care of the following minor revision described below.*

Response: We deeply appreciate your valuable suggestions and time spent reviewing our manuscript.

Specific comments:

*1. Page 5: Why was Sansha chosen as the research area? How does the intensity of marine biological activities there compare with those in other tropical seas?*

Response: We chose Sansha because it is the only tropical sea area that allows us to conduct one-year field observations.

*2. Page 7: the authors mention that all OSs were undetectable in the blank samples. However, the specific handling procedures for the blank samples were not specified. It is recommended to provide more details on the quality control of the blank samples.*

Response: We greatly appreciate your comments. We have updated the relevant content.

Lines 124–125: A blank filter sample was collected on a monthly basis using the same air sampler (without operation) as referenced above.

Lines 147–148: All OSs were undetectable in the blank samples with the same measurement methodology as mentioned above.

*3. Page 9: Why did you choose the forward mode with the thermodynamically metastable state when using ISORROPIA-II to predict the pH value?*

Response: The model was run in the forward mode for metastable aerosols, which was shown to yield more accurate aerosol pH predictions than reverse-mode calculations using only aerosol data input (Wang et al., 2021; Guo et al., 2015; Hennigan et al., 2015).

More descriptions have been added to the SI.

S4. Prediction of pH: …The model was run in forward mode for metastable aerosols, yielding more accurate aerosol pH predictions than reverse-mode calculations using only aerosol data input (Wang et al., 2021; Guo et al., 2015; Hennigan et al., 2015) …

*4. Page 10: Were the OSs and NOCs detected in the study reported in other marine aerosols? Are their abundances geographically specific?*

Response: The occurrence of OSs in marine aerosols has been reported (**Table S4** in the SI). The abundance of oceanic OS exhibits spatial variation, although there is limited research on marine aerosol OSs. However, there were not as many types of marine aerosol OSs reported previously as in this study, especially the lack of reports on the $OS_a$ group. This also indicates that our work is sufficiently novel and unique.

The determination of NOC molecular abundance is tightly associated with different instruments, methodologies, and operators. Thus, in this study, the NOC data reported for comparison were derived from our research group, and all NOC data showing in Figure 1 were obtained with the same instruments, methodology, and operators.

Lines 164–166: …an intercomparison of the relative abundance of compounds identified with the same analytical approach and instrument by the same person was performed in the present study (Ma et al., 2025)…

Lines 239–240: …The NOC data were identified using the identical analysis methodology (Ma et al., 2025)…

*5. Page 11: Figure 1 compares the OSs and NOCs data of different cities, but does not specify whether the sampling times and analysis methods of these data are consistent. It is suggested that a description of the comparability of the methods be added.*

Response: Table S4 (the data in Figure 1 is presented in bold) in the SI shows the specific sampling times reported in different studies. Due to the fact that OSs are quantitatively analyzed and the analytical methods (LC-MS method) are generally similar, comparisons between different studies are feasible. The determination of NOC molecular abundance is tightly associated with different instruments, methodologies, and operators. Thus, in this study, the NOC data reported for comparison were derived from our research group, and all NOC data showing in Figure 1 were obtained with the same instruments, methodology, and operators (Lines 164–166 and 239–240).

*6. Page 14: Is it possible that the formation of NOCs in low $NH_4^+$ environments occurs through other pathways (such as reactions involving organic amines)?*

Response: The abundance of organic amines is usually about 100–1000 times lower than that of ammonia. In this study site, the abundance of atmospheric ammonia (/ammonium) is quite low (up to ten times lower relative to other cities), therefore, ammonia and organic amines are expected to be constraining factors in the formation of NOCs.

Some updated descriptions have been added to the revised manuscript.

Lines 287–292: …Presumably, the significantly lower levels (up to ten times lower) of aerosol $NH_4^+$ (**Table S1**) and gaseous $NH_3$ in the Sansha area relative to other cities (Ma et

al., 2025; Pan et al., 2018; Dong et al., 2023) may be one of the important factors constraining Re-NOC formation. The insignificant correlation between CHON+ abundance and $NH_4^+$ concentration ($P > 0.05$) also partially supports the above inference…

*7. Page 20: the authors mention that the above results suggest that ship emissions were not the primary factor in controlling the abundance of aerosol OSs and NOCs in the Sansha area. It is suggested to provide specific percentages to illustrate this point, and to elaborate on the results shown in Figure 4.*

Response: We cannot provide a specific percentage to quantify the impact of ship emissions. Because this conclusion is a reasonable speculation we made based on the results of data analysis. Briefly, if the impact of ship emissions on the formation of OSs and NOCs is significant, then the changes in OS and NOC abundances are at least consistent with ship VOC emissions. However, we did not observe this phenomenon.

*8. Page 21: the authors mention that a Mantel test correlation analysis was conducted to further investigate the impact of different factors on the formation of OSs and NOCs in the Sansha area. However, the criteria for variable selection (such as why Na⁺ was chosen instead of other ions) were not explained. It is suggested to provide the basis for variable selection.*

Response: Sodium ions and Chl a have been identified as important indicators of marine sources (Wang et al., 2023; Xiao et al., 2017). Thus, in this study, we prioritized sodium

ions and Chl a as indicators of marine sources.

We have updated the relevant content in the revised manuscript.

Lines 453–455: …Sodium ions and Chl a have been identified as important indicators of marine sources (Wang et al., 2023; Xiao et al., 2017)…

**Once again, we deeply appreciate the time and effort you've spent in reviewing our manuscript.**

**References**

Dong, J., Li, B., Li, Y., Zhou, R., Gan, C., Zhao, Y., Liu, R., Yang, Y., Wang, T., and Liao, H.: Atmospheric ammonia in China: Long-term spatiotemporal variation, urban-rural gradient, and influencing factors, Science of The Total Environment, 883, 163733, https://doi.org/10.1016/j.scitotenv.2023.163733, 2023.

Guo, H. Y., Xu, L., Bougiatioti, A., Cerully, K. M., Capps, S. L., Hite Jr, J., Carlton, A., Lee, S. H., Bergin, M., and Ng, N.: Fine-particle water and pH in the southeastern United States, Atmos. Chem. Phys., 15, 5211-5228. https://doi.org/5210.5194/acp-5215-5211-2015, 2015.

Hennigan, C., Izumi, J., Sullivan, A., Weber, R., and Nenes, A.: A critical evaluation of proxy methods used to estimate the acidity of atmospheric particles, Atmos. Chem. Phys., 15, 2775-2790. https://doi.org/2710.5194/acp-2715-2775-2015, 2015.

Ma, Y. J., Xu, Y., Yang, T., Gui, L., Xiao, H. W., Xiao, H., and Xiao, H. Y.: The critical

role of aqueous-phase processes in aromatic-derived nitrogen-containing organic aerosol formation in cities with different energy consumption patterns, Atmos. Chem. Phys., 25, 2763-2780, 10.5194/acp-25-2763-2025, 2025.

Pan, Y., Tian, S., Zhao, Y., Zhang, L., Zhu, X., Gao, J., Huang, W., Zhou, Y., Song, Y., Zhang, Q., and Wang, Y.: Identifying Ammonia Hotspots in China Using a National Observation Network, Environmental Science & Technology, 52, 3926-3934, 10.1021/acs.est.7b05235, 2018.

Wang, Y., Zhao, Y., Wang, Y., Yu, J. Z., Shao, J., Liu, P., Zhu, W., Cheng, Z., Li, Z., Yan, N., and Xiao, H.: Organosulfates in atmospheric aerosols in Shanghai, China: seasonal and interannual variability, origin, and formation mechanisms, Atmos. Chem. Phys., 21, 2959-2980, 10.5194/acp-21-2959-2021, 2021.

Wang, Y., Zhang, Y., Li, W., Wu, G., Qi, Y., Li, S., Zhu, W., Yu, J. Z., Yu, X., Zhang, H.-H., Sun, J., Wang, W., Sheng, L., Yao, X., Gao, H., Huang, C., Ma, Y., and Zhou, Y.: Important Roles and Formation of Atmospheric Organosulfates in Marine Organic Aerosols: Influence of Phytoplankton Emissions and Anthropogenic Pollutants, Environ. Sci. Technol., 57, 10284-10294, 10.1021/acs.est.3c01422, 2023.

Xiao, H. W., Xiao, H. Y., Luo, L., Shen, C. Y., Long, A. M., Chen, L., Long, Z. H., and Li, D. N.: Atmospheric aerosol compositions over the South China Sea: temporal variability and source apportionment, Atmos. Chem. Phys., 17, 3199-3214, 2017.

---

## Author Comment (AC3)

**General.**

We would like to appreciate the referee for providing the valuable comments to improve the manuscript. We have revised our manuscript by fully taking the editor's comments into account. Responses to specific comments are described below. All the changes made and appeared in the revised text are shown in red. All detailed answers to comments are displayed in blue.

**Comments of Referee #3 and our responses to them**

*This study reports the molecular composition and characteristics of marine organic aerosols in Sansha, South China Sea, and especially focuses on the OSs and NOCs. The data is very informative and valuable. The following comments need to be addressed before being published on ACP.*

Response: We deeply appreciate your valuable suggestions and time spent reviewing our manuscript.

General comments:

*1. This work highlights the contribution of marine emissions to aromatic/aliphatic organic sulfur and nitrogen aerosols. My major concern is the potential influence of biomass burning on the aromatic/aliphatic sulfur or nitrogen, or other organic compounds in marine aerosols over the South China Sea.*

*Many shipboard cruise observations have suggested the obvious impacts of biomass burning emissions on the marine organic aerosol formation over the South China Sea, including studies from Chinese researchers. The authors also report that higher*

*concentrations of OSs and NOCs and more fire pots were observed during December to March. Biomass burning has been proved to be one. Thus, more solid and detailed evidence should be provided to exclude the possible impacts of biomass burning emissions on the formation of sulfur/nitrogen organic aerosols.*

*The K+ concentrations in June-August are much higher than during other months. I may suggest analyzing the data during Aug. 20—May (without the data in Jun-Aug) and during Jun-Aug separately for the correlation analysis in Fig. 5 or the PCA analysis in Fig. 6. It should also be noted that previous studies have reported that the K+ in biomass burning aerosols would decrease rapidly during long-range transport in the atmosphere.*

Response: We are very sorry for the confusion caused by our insufficient discussions. We carried out the correlation analysis between potassium ions and the organic compounds under investigation in phases in accordance with your instructions (Figure 1). The results showed a weak correlation between potassium ions and most organic components. In addition, we have demonstrated from another perspective the weak impact of biomass burning pollutants transported over long distances on the aerosol OSs and NOCs in the Sansha area.

The analysis of air mass back trajectories shows that although biomass combustion activity is more intense in inland areas during winter and spring, the main air masses arriving at study stie have not passed through the combustion intensive areas. Furthermore, if we assume that biomass burning in inland or coastal areas transferred OSs to Sansha, this will only lead to an increase in $OS_a$ (anthropogenic origin) abundances, rather than a highly consistent increase in $OS_i$, $OS_m$ (biological origin), and $OS_a$ abundances. In particular, OSs showed significant correlations ($r > 0.6$, $P < 0.01$) with local $O_3$ levels.

The above results suggest that aerosol OSs in the Sansha area may be mainly formed locally

[Figure]

Figure 1. Correlation analysis between nss-$K^+$ and investigated organic compounds for data during (a) Aug. 20—May and (b) Jun-Aug.

We have added more descriptions in the revised manuscript.

Lines 344–348: …these OSs showed significant correlations ($r > 0.6$, $P < 0.01$) with local $O_3$ levels. The above results suggest that aerosol OSs in the Sansha area may be mainly formed locally and are tightly associated with precursor emission levels (e.g., abovementioned phytoplankton emissions)...

Lines 409–412: Even if we assume that biomass burning in inland or coastal areas transported OSs to the Sansha region, it would only lead to a significant increase in $OS_a$ abundances but not induce a synchronous increase in $OS_i$, $OS_m$, and $OS_a$ abundances…

*2. There have been some studies on the seasonal variations of seawater isoprene or other*

*VOCs. I may suggest the authors combining the observation results on the seawater VOCs to validate the seasonal variations of satellite-derived Chl-a and calculated isoprene concentration reported here. Does the data here follow the same seasonal variation trend as reported in previous studies?*

Response: We greatly appreciate your comments. The variation patterns among sea surface temperatures, seawater isoprene levels, and Chlorophyll-a obtained from empirical formulas or satellite data presented here are consistent with actual observation results in the South China Sea (Zhai et al., 2018).

Lines 326–329: The variation patterns among SST, seawater isoprene levels, and Chlorophyll-a obtained from empirical formulas or satellite data presented here are consistent with actual observation results in the South China Sea (Zhai et al., 2018).

Specific comments:

*1. Lines 287-291: The authors argue that the much lower NH4+ aerosols lead to the limited formation of Re-NOCs. Is the gaseous NH3 lower in Sansha than in other cities? Reactions between carbonyl compounds and gaseous NH3 could also form CHON+. Considering the higher air temperature in Sansha, the fraction of NH3 in the gas phase could be much higher in Sansha than in other cities.*

Response: The gaseous $NH_3$ level in the South China Sea region is usually lower than that in investigated inland cities (Dong et al., 2023; Pan et al., 2018). More references have been cited in the revised manuscript (Lines 287–290).

Some updated descriptions have been added to the revised manuscript.

Lines 287–290: …Presumably, the significantly lower levels (up to ten times lower) of aerosol $NH_4^+$ (**Table S1**) and gaseous $NH_3$ in the Sansha area relative to other cities (Ma et al., 2025; Pan et al., 2018; Dong et al., 2023) may be one of the important factors constraining Re-NOC formation.…

*2. The authors compare the NOCs compositions in Sansha and other sites. However, different analysis MS instruments or measurement parameters were used in different studies. This might be the primary reason for the different NOC compositions.*

Response: The NOC data used for comparison were derived from our research group (Ma et al., 2025), and all NOC data showing in Figure 1 were obtained with the same instruments, methodology, and operators.

Lines 164–166: …an intercomparison of the relative abundance of compounds identified with the same analytical approach and instrument by the same person was performed in the present study (Ma et al., 2025)…

Lines 239–240: …The NOC data were identified using the identical analysis methodology (Ma et al., 2025)…

**Once again, we deeply appreciate the time and effort you've spent in reviewing our manuscript.**

**References**

Dong, J., Li, B., Li, Y., Zhou, R., Gan, C., Zhao, Y., Liu, R., Yang, Y., Wang, T., and Liao, H.: Atmospheric ammonia in China: Long-term spatiotemporal variation, urban-rural gradient, and influencing factors, Science of The Total Environment, 883, 163733, https://doi.org/10.1016/j.scitotenv.2023.163733, 2023.

Ma, Y. J., Xu, Y., Yang, T., Gui, L., Xiao, H. W., Xiao, H., and Xiao, H. Y.: The critical role of aqueous-phase processes in aromatic-derived nitrogen-containing organic aerosol formation in cities with different energy consumption patterns, Atmos. Chem. Phys., 25, 2763-2780, 10.5194/acp-25-2763-2025, 2025.

Pan, Y., Tian, S., Zhao, Y., Zhang, L., Zhu, X., Gao, J., Huang, W., Zhou, Y., Song, Y., Zhang, Q., and Wang, Y.: Identifying Ammonia Hotspots in China Using a National Observation Network, Environmental Science & Technology, 52, 3926-3934, 10.1021/acs.est.7b05235, 2018.

Zhai, X., Zhang, H.-H., Yang, G.-P., Li, J.-L., and Yuan, D.: Distribution and sea-air fluxes of biogenic gases and relationships with phytoplankton and nutrients in the central basin of the South China Sea during summer, Mar. Chem., 200, 33-44, https://doi.org/10.1016/j.marchem.2018.01.009, 2018.

---

## Author Comment (AC4)

**General.**

We would like to appreciate the referee for providing the valuable comments to improve the manuscript. We have revised our manuscript by fully taking the editor's comments into account. Responses to specific comments are described below. All the changes made and appeared in the revised text are shown in red. All detailed answers to comments are displayed in blue.

**Comments of Referee #4 and our responses to them**

*This manuscript presents an observational study on organosulfates (OSs) and nitrogen-containing organic compounds (NOCs) in fine particulate matter in an island in the South China Sea. The source of marine emissions has been emphasized though other source contribution of long-range transport could still be found. This study highlights the contribution of marine biogenic emissions for OSs and NOCs in the oceanic area. I would recommend it to be published in ACP after moderate revision. My comments/suggestions are listed below.*

Response: We deeply appreciate your valuable suggestions and time spent reviewing our manuscript.

General comments:

*1. Line 254-255, Page 12: As the proportions of OSa in the total OSs are higher at both inland and ocean sites, how can we differentiate between the anthropogenic and oceanic sources of OSa based on its percentage in aerosol samples?*

Response: The mass fractions of $OS_a$ (aromatic and aliphatic SOA) in the total OSs in

inland China were smaller than that in Sansha Island (a region with minimal anthropogenic pollution), although the $OS_a$ concentration was lower in Sansha Island (Figure 1a and Table S4). The attribution of this phenomenon to the long-range transport of pollutants from coastal cities in China may not be a reasonable assumption. This is due to the fact that the transmission effect was incapable of inducing an increase in the proportion of $OS_a$ from coastal cities with developed industries (e.g., Shanghai and Guangzhou) to Sansha Island (see section 3.3 for further evidence). In addition, the SOA markers we detected showed a significant positive correlation with ozone ($P$<0.01, Figure 5), indicating that these SOA markers were formed locally (Lines 245–264).

*2. Line 295-297, Page 14: The assertion that atmospheric oxidation capacity does not constrain OS formation is contradicted by the significant correlation between OSs and O3 mentioned in 3.2.*

Response: We are very sorry for the confusion caused by our description. Here, what we want to express is that atmospheric oxidation capacity is not a limiting factor for OS formation at the site of this study. We have rephrased the sentence in the revised draft.

Lines 306–307: …sulfate availability and atmospheric oxidation capacity would not be limiting factors for OS formation…

*3. Ling 330, Page 16: Given that both marine biological activities and anthropogenic sources contribute to atmospheric sulfate, what are the non-sea salt and sea salt proportions of the total sulfate in aerosol samples, and are there any differences in*

*correlations of OSs versus these sulfate fractions?*

Response: If sea salt sulfates are defined as 0.060 times the sodium ion concentration based on previous report (Chen et al., 2010), the calculated average proportion of non-sea -salt sulfates is approximately 93% of the total sulfates. However, this method does not seem to be suitable for calculating the actual concentration of sea-salt sulfates. This is because the sulfates in the $PM_{2.5}$ sample are mainly formed through secondary processes associated with precursors emitted from the ocean (e.g., DMS transformation), and sea-salt sulfates and sodium should be mainly distributed in coarse particles. Therefore, we cannot accurately distinguish the ratio of sea-salt and non-sea-salt sulfates here. However, this study can at least demonstrate that abundant marine aerosol sulfates are not a limiting factor for OS formation by comparing observation cases of coastal cities and Sansha.

The following content is what we want to emphasize

Lines 345–349: OSs showed significant correlations ($r > 0.6$, $P < 0.01$) with local $O_3$ levels. The above results suggest that aerosol OSs in the Sansha area may be mainly formed locally and are tightly associated with precursor emission levels (e.g., abovementioned phytoplankton emissions) rather than sulfates.

*4. Line 388-389, Page 19: How to explain the coincidence between the maximum of biomass burning intensity and elevated levels of OSs and aromatic NOCs during winter and spring months, assuming the SCS atmosphere is unaffected by air mass long-range transport?*

Response: The analysis of air mass backward trajectories shows that although biomass

combustion activity is more intense in inland areas during winter and spring, the main air masses arriving at study stie have not passed through the combustion intensive areas. Furthermore, if we assume that biomass burning in inland or coastal areas transferred OSs to Sansha, this will only lead to an increase in $OS_a$ abundances, rather than a highly consistent increase in $OS_i$, $OS_m$ (biological origin), and $OS_a$ abundances.

We have added more descriptions in the revised manuscript.

Lines 410–413: Even if we assume that biomass burning in inland or coastal areas transported OSs to the Sansha region, it would only lead to a significant increase in $OS_a$ abundances but not induce a synchronous increase in $OS_i$, $OS_m$, and $OS_a$ abundances…

5. *Line 392-394, Page 19: The inconsistent variation trends between nss-K+ and the surrounding continental fire point density alone cannot conclusively demonstrate that the SCS atmosphere is not impacted by air mass long-range transport. Are there additional anthropogenic or terrestrial tracer concentration patterns strengthening this assumption?*

Response: If we assume that biomass burning in inland or coastal areas transferred OSs to Sansha, this will only lead to an increase in $OS_a$ abundances, rather than a highly consistent increase in $OS_i$, $OS_m$ (biological origin), and $OS_a$ abundances (Lines 410–413).

6. *Line 415-417, Page 20: Does the inverse relationship between isoprene/monoterpenes and OSs/aromatic NOCs contradict the previous allusion that the predominance of isoprene and monoterpenes-derived NOCs over aromatic NOCs results from marine*

*emissions of isoprene and monoterpenes?*

Response: Here we refer to the emissions of isoprene and monoterpenes from ships. Their lack of significant positive correlation with OSs and aromatic NOCs indicates that ship emissions are not the main factor driving the temporal changes in OSs and NOCs in the Sansha area.

*7. Line 439-441, Page 22: Could the author provide the exact Mantel's r and p value from the Mantel test, since the threshold of >4 is too broad for meaningful interpretation?*

Response: The code we used cannot directly input all specific Mantel's r and p values. Thus, we performed PCA analysis and correlation analysis between specific markers ($C_6H_5O_4S^-$ and $C_7H_7O_4S^-$) and key parameters (**Figure 6**) to further support our inference.

*8. Line 487, Page 25: How can Ca2+ indicate marine sources given its negative correlation with Na+ and Cl- in Figure 5?*

Response: We have removed the inappropriate statements. Indeed, calcium ions, as mineral ions that can be transported over long distances and are mostly present in coarse particles, should not be hastily determined for their oceanic origin.

*9. Line 502, Page 25: Did these two aromatic OSs dominate in the total aromatic OSs? How do other aromatic species correlate with these marine emission indicators?*

Response: On average, $C_6H_5O_4S^-$ and $C_7H_7O_4S^-$ accounted for only about 6% of the total

aromatic $OS_s$ (**Table S3**). The total aromatic OSs were significantly correlated with the marine emission indicators. We chose these two SOA markers because their structures have been resolved and we have a clear understanding of their precursors.

The detailed discussion is as follows:

Lines 508–518 : The release of benzene and toluene from marine organisms has been suggested to be important precursors for marine SOA (Wohl et al., 2023). It has been established that the reaction of benzene and toluene with sulfate radicals in the aqueous phase can lead to the formation of two aromatic OSs, namely $C_6H_5O_4S^-$ and $C_7H_7O_4S^-$ (Huang et al., 2020). We detected both aromatic OSs in aerosols collected in the Sansha area, the structures of which were shown in **Figure 6b**. Moreover, we found that these two benzene- and toluene-derived OSs showed significant positive correlations with multiple indicators of marine emissions (e.g., Chl a, surface seawater isoprene, $Na^+$, $OS_i$, and $OS_m$) (**Figure 6b**). This further corroborates the important effect of marine emissions on the formation of aromatic-derived SOA in the Sansha area…

**Once again, we deeply appreciate the time and effort you've spent in reviewing our manuscript.**

**References**

Chen, H. Y., Chen, L. D., Chiang, Z. Y., Hung, C. C., Lin, F. J., Chou, W. C., Gong, G. C., and Wen, L. S.: Size fractionation and molecular composition of water‑soluble

inorganic and organic nitrogen in aerosols of a coastal environment, J. Geophys. Res.: Atmos., 115, doi: 10.1029/2010JD014157., 2010.

Huang, L., Liu, T., and Grassian, V. H.: Radical-Initiated Formation of Aromatic Organosulfates and Sulfonates in the Aqueous Phase, Environmental Science & Technology, 54, 11857-11864, 10.1021/acs.est.0c05644, 2020.

Wohl, C., Li, Q., Cuevas, C. A., Fernandez, R. P., Yang, M., Saiz-Lopez, A., and Simó, R.: Marine biogenic emissions of benzene and toluene and their contribution to secondary organic aerosols over the polar oceans, Science Advances, 9, eadd9031, doi:10.1126/sciadv.add9031, 2023.

---

## Author Response (AR2)

**General.**

We would like to appreciate the editor for providing the valuable comments to improve the manuscript. We have revised our manuscript by fully taking the editor's comments into account. Responses to specific comments are described below. All the changes made and appeared in the revised text are shown in red. All detailed answers to comments are displayed in blue.

**Comments of editor and our responses to them**

The authors have addressed the reviewers' questions and concerns comprehensively. However, the abstract and concluding sections are missing some elements required by the journal. I ask authors to revise those sections according to the ACP Guidelines for the title, abstract, and concluding section:

https://www.atmospheric-chemistry-and-physics.net/policies/guidelines\_for\_authors.html.

Response: We sincerely appreciate your careful review of the manuscript. We have updated the relevant content.

Lines 24–26: ...Marine atmospheric organic aerosols play a pivotal role in regulating global climate dynamics and influencing marine biogeochemical cycles. Compared to the extensive research on marine isoprene-derived secondary organic aerosols (SOA), the origins of marine aromatic and aliphatic organic aerosols remain elusive...

Lines 521–543: Based on our current understanding, this study represents the inaugural instance of simultaneous comprehensive characterization of OSs and NOCs (in both ESI+ and ESI– modes) in PM2.5 in tropical marine areas, particularly in the Sansha area with minimal anthropogenic pollution. The significant correlation (r = 0.61-0.71, P < 0.01)

between estimated surface seawater isoprene and OSi and OSm indicated the important role of marine biological emissions in the formation of marine aerosol BSOA. Integrated analysis of air mass backward trajectories, Mantel tests, PCA, and specific molecular tracers (e.g., C6H5O4S- and C7H7O4S-) revealed that precursors originating from the marine environment may substantially contribute to the formation of aliphatic and aromatic OSs and NOCs in the Sansha area. We concluded that the emissions of marine organisms can contribute to the formation of both typical BSOA (i.e., isoprene and monoterpenes-derived species) and aliphatic- and aromatic-derived SOA in this sea area. A recent study in the Yellow Sea of China has reported that marine phytoplankton or microorganisms can contribute to the formation of marine aerosol OSi and OSm (Wang et al., 2023). However, the available information regarding the origins of aliphatic and aromatic OSs and NOCs in marine aerosols was previously inadequate. It has often been assumed that aliphatic and aromatic pollutants in marine aerosols are predominantly derived from long-range atmospheric transport from land, with less consideration given to local marine-derived sources or secondary formation (Zhou et al., 2023; Sun et al., 2023; Hansen et al., 2014). Thus, this is the first field observation case to demonstrate that marine organisms may also provide important aliphatic or aromatic precursors for the formation of aliphatic and aromatic OSs and NOCs.

Hansen, A. M. K., Kristensen, K., Nguyen, Q. T., Zare, A., Cozzi, F., Nøjgaard, J. K., Skov,
H., Brandt, J., Christensen, J. H., Ström, J., Tunved, P., Krejci, R., and Glasius, M.:
Organosulfates and organic acids in Arctic aerosols: speciation, annual variation and concentration levels, Atmos. Chem. Phys., 14, 7807-7823, 10.5194/acp-14-7807-2014, 2014.

- Sun, Q., Liang, B., Cai, M., Zhang, Y., Ou, H., Ni, X., Sun, X., Han, B., Deng, X., Zhou, S., and Zhao, J.: Cruise observation of the marine atmosphere and ship emissions in South China Sea: Aerosol composition, sources, and the aging process, Environmental pollution, 316, 120539, <a href="https://doi.org/10.1016/j.envpol.2022.120539">https://doi.org/10.1016/j.envpol.2022.120539</a>, 2023.
- Wang, Y., Zhang, Y., Li, W., Wu, G., Qi, Y., Li, S., Zhu, W., Yu, J. Z., Yu, X., Zhang, H.-H.,
  Sun, J., Wang, W., Sheng, L., Yao, X., Gao, H., Huang, C., Ma, Y., and Zhou, Y.:
  Important Roles and Formation of Atmospheric Organosulfates in Marine Organic
  Aerosols: Influence of Phytoplankton Emissions and Anthropogenic Pollutants,
  Environ. Sci. Technol., 57, 10284-10294, 10.1021/acs.est.3c01422, 2023.
- Zhou, S., Guo, F., Chao, C.-Y., Yoon, S., Alvarez, S. L., Shrestha, S., Flynn, J. H., III, Usenko, S., Sheesley, R. J., and Griffin, R. J.: Marine Submicron Aerosols from the Gulf of Mexico: Polluted and Acidic with Rapid Production of Sulfate and Organosulfates, Environmental Science & Technology, 57, 5149-5159, 10.1021/acs.est.2c05469, 2023.

---

## Author Response (AR3)

**Remarks from the preceding review file validation**

Please ensure that the colour schemes used in your maps and charts allow readers with colour vision deficiencies to correctly interpret your findings. Please check your figures using the Coblis — Color Blindness Simulator (https://www.color-blindness.com/cobliscolor-blindness-simulator/) and revise the colour schemes accordingly. --> Fig. 2 and S3. Response: We sincerely appreciate your careful review of the manuscript. We have checked the data or trend lines in Fig. 2 and S3 through Color Blindness Simulator, which can be distinguished by the depth or brightness of the colors.